# Effects of Lifestyle Modification Interventions to Prevent and Manage Child and Adolescent Obesity: A Systematic Review and Meta-Analysis

**DOI:** 10.3390/nu12082208

**Published:** 2020-07-24

**Authors:** Rehana A. Salam, Zahra A. Padhani, Jai K. Das, Amina Y. Shaikh, Zahra Hoodbhoy, Sarah Masroor Jeelani, Zohra S. Lassi, Zulfiqar A. Bhutta

**Affiliations:** 1Division of Women and Child Health, Aga Khan University Hospital, Karachi 74800, Pakistan; rehana.salam@aku.edu (R.A.S.); zahra.feroz@aku.edu (Z.A.P.); jai.das@aku.edu (J.K.D.); amina.shaikh@hotmail.com (A.Y.S.); zahra.hoodbhoy@aku.edu (Z.H.); 2Department of Trauma & Orthopaedics, Macclesfield District General Hospital, East Cheshire NHS Trust Victoria Rd, Macclesfield SK10 3BL, UK; saarc93@yahoo.com; 3Robinson Research Institute, University of Adelaide, Adelaide, SA 5005, Australia; zohra.lassi@adelaide.edu.au; 4Centre for Global Child Health, the Hospital for Sick Children, Toronto, ON M5G 1X8, Canada

**Keywords:** obesity, interventions, children, adolescents

## Abstract

The objective of this review was to assess the impact of lifestyle interventions (including dietary interventions, physical activity, behavioral therapy, or any combination of these interventions) to prevent and manage childhood and adolescent obesity. We conducted a comprehensive literature search across various databases and grey literature without any restrictions on publication, language, or publication status until February 2020. We included randomized controlled trials and quasi-experimental studies from both high income countries (HIC) and low-middle-income countries (LMICs). Participants were children and adolescents from 0 to 19 years of age. Studies conducted among hospitalized children and children with any pre-existing health conditions were excluded from this review. A total of 654 studies (1160 papers) that met the inclusion criteria were included in this review. A total of 359 studies targeted obesity prevention, 280 studies targeted obesity management, while 15 studies targeted both prevention and management. The majority of the studies (81%) were conducted in HICs, 10% of studies were conducted in upper middle income countries, while only 2% of the studies were conducted in LMICs. The most common setting for these interventions were communities and school settings. Evidence for the prevention of obesity among children and adolescents suggests that a combination of diet and exercise might reduce the BMI *z*-score (MD: −0.12; 95% CI: −0.18 to −0.06; 32 studies; 33,039 participants; I^2^ 93%; low quality evidence), body mass index (BMI) by 0.41 kg/m^2^ (MD: −0.41 kg/m^2^; 95% CI: −0.60 to −0.21; 35 studies; 47,499 participants; I^2^ 98%; low quality evidence), and body weight (MD: −1.59; 95% CI: −2.95 to −0.23; 17 studies; 35,023 participants; I^2^ 100%; low quality evidence). Behavioral therapy alone (MD: −0.07; 95% CI: −0.14 to −0.00; 19 studies; 8569 participants; I^2^ 76%; low quality evidence) and a combination of exercise and behavioral therapy (MD: −0.08; 95% CI: −0.16 to −0.00; 9 studies; 7334 participants; I^2^ 74%; low quality evidence) and diet in combination with exercise and behavioral therapy (MD: −0.13; 95% CI: −0.25 to −0.01; 5 studies; 1806 participants; I^2^ 62%; low quality evidence) might reduce BMI *z*-score when compared to the control group. Evidence for obesity management suggests that exercise only interventions probably reduce BMI *z*-score (MD: −0.13; 95% CI: −0.20 to −0.06; 12 studies; 1084 participants; I^2^ 0%; moderate quality evidence), and might reduce BMI (MD: −0.88; 95% CI: −1.265 to −0.50; 34 studies; 3846 participants; I^2^ 72%) and body weight (MD: −3.01; 95% CI: −5.56 to −0.47; 16 studies; 1701 participants; I^2^ 78%; low quality evidence) when compared to the control group. and the exercise along with behavioral therapy interventions (MD: −0.08; 95% CI: −0.16 to −0.00; 8 studies; 466 participants; I^2^ 49%; moderate quality evidence), diet along with behavioral therapy interventions (MD: −0.16; 95% CI: −0.26 to −0.07; 4 studies; 329 participants; I^2^ 0%; moderate quality evidence), and combination of diet, exercise and behavioral therapy (MD: −0.09; 95% CI: −0.14 to −0.05; 13 studies; 2995 participants; I^2^ 12%; moderate quality evidence) also probably decreases BMI *z*-score when compared to the control group. The existing evidence is most favorable for a combination of interventions, such as diet along with exercise and exercise along with behavioral therapy for obesity prevention and exercise alone, diet along with exercise, diet along with behavioral therapy, and a combination of diet, exercise, and behavioral therapy for obesity management. Despite the growing obesity epidemic in LMICs, there is a significant dearth of obesity prevention and management studies from these regions.

## 1. Introduction

Obesity is a major public health crisis for children and adults across the world [1]. The Global Nutrition Report 2019 highlights that about 40.1 million children globally are overweight and at the same time, overweight, and obesity are increasing rapidly in nearly every country in the world, with no signs of slowing [2]. It is estimated that in 2015, approximately 10% of children and adults globally were obese [3]. The non-communicable disease risk factor collaboration reported that the global prevalence of obesity in boys increased from 0.7% in 1975 to 5.6% in 2016, while in girls the increase was from 0.9% to 7.8% in boys during the same time duration [4]. Data from the National Health and Nutrition Examination Survey (NHANES) reported that in United States, obesity in boys aged 2–5 years had a steep incline from 1999–2016, while there was an increase in overweight girls aged 16–19 years [5]. Similar trends were noted in the data obtained from the United Kingdom, where the obesity epidemic continued to rise in the adolescent population [6]. Low middle income countries (LMICs) are plagued with the “double burden of disease” epidemic [7]. Popkin et al. have demonstrated a significant increase in body mass index (BMI) and waist circumference in individuals from LMICs including Asia and Africa [8]. The prevalence of overweight and obesity in adolescents in LMICs was 15% and 6% respectively [9]. This was particularly noted in Asian regions, Latin America, and Africa [4]. Childhood obesity is known to be associated with a myriad of morbidities, such as atherosclerosis, hypertension, diabetes, metabolic syndrome etc. [10]. The rising obesity epidemic and its associated complications led to the World Health Organization Commission on Ending Childhood Obesity recommending three strategic areas for action [11]. These include prevention through heath education, treatment for childhood obesity, and advocacy regarding healthier environments [11].

Childhood obesity interventions include lifestyle modifications, pharmacological, and surgical interventions [12]. For the purpose of this review, we restricted the interventions to lifestyle modifications only, including dietary interventions, physical activity interventions, and behavioral therapy. Lifestyle modification with increased intake of fruits and vegetables along with recommended moderate physical activity form the mainstay for primary prevention of childhood obesity [13]. Pineda et al. have demonstrated that changes in the school food environment, such as a ban on sugar sweetened beverages and increased availability of fruits and vegetables has led to a significant decrease in obesity prevalence [14]. Nutrition education, as well as school meal policy changes, have been the driving forces behind the reduction in obesity reported in these studies [14]. Physical activity interventions for childhood obesity include promotion of exercise and reduction in sedentary behavior. School-based exercise interventions have been shown to be associated with a lower body mass index (BMI) in children as compared to their inactive counterparts [15]. Reducing sedentary behavior by restricting television and computer time is known to prevent excessive weight gain in children [16]. These interventions ensure that there is no excessive caloric gain thus preventing obesity. Behavioral interventions, such as cognitive behavioral therapy (CBT) are known to be effective in addressing several health conditions including adult and childhood obesity [17]. This technique encourages participants to self-regulate their diet and activity routines to improve their weight management. These interventions are largely implemented in school or community settings, where children and adolescents are available and follow up is convenient over a long period [18,19,20,21,22]. Schools tend to be an ideal location for intervention due to the facilities available and easy access to this particular age group; they also offer time slots and equipment facility to encourage adequate physical activity [23,24]. Other delivery platforms include after-school settings, sports clubs, or within family settings [24].

Multiple reviews and meta-analyses exist evaluating the efficacy of interventions to prevent and manage obesity in children and adolescent [12,25,26,27,28]. With the variety of interventions and the ever increasing number of reviews, it is difficult to generate conclusions regarding which interventions are relatively more effective, compared to others for preventing and managing obesity in this age group [24]. Moreover, the results of the existing meta-analyses do not always point in the same direction. A recent meta-synthesis has attempted moderator analyses to explain why some interventions are more effective than others; however, it is limited to a few factors [24]. Furthermore, the majority of the existing systematic reviews have restricted their inclusions to randomized controlled trials (RCTs) alone and focused too much on effectiveness aspect without focusing on the various contextual factors that might potentially impact the effectiveness of these interventions. Although RCTs are considered to be the gold standard when evaluating effectiveness, complementing RCT data with observational studies is sometimes imperative when evaluating complex lifestyle and behavioral interventions [29,30,31,32,33]. Behavioral interventions are complex and influenced by various individual and environmental factors that could potentially impact the uptake and effectiveness of these interventions.

This review aims to summarize up-to-date evidence for both children and adolescents. In addition, we also assessed the studies included in our review with the lens of the PROGRESS framework (place, race, occupation, gender, religion, education, socioeconomic status, and social status). The findings from this review will provide a basis to guide public health program planners to adapt these interventions (alone or in combination) based on identifying factors that may affect how some groups engage with the intervention or the method of implementation along with an insight to the public health researchers on whether outcomes differed by relevant socio-demographic characteristics and whether the intervention was effective for disadvantaged groups. The protocol for this review was published with the Campbell Collaboration at https://onlinelibrary.wiley.com/doi.org/10.1002/CL2.192.

## 2. Materials and Methods

### 2.1. Objectives

The objective of this review was to assess the impact of lifestyle interventions (including dietary interventions, physical activity, behavioral therapy, or any combination of these interventions) along with the contextual factors to prevent and manage childhood and adolescent obesity.

### 2.2. Types of Studies and Participants

Keeping in mind the objective of this work, we included primary studies using randomized control trials (individual and cluster) and quasi-experimental studies from both high income countries (HIC) and LMICs. Participants were children and adolescents aged 0 to 19 years. Studies conducted among hospitalized children were excluded from this review. We also excluded children with any pre-existing health conditions (e.g., diabetes and kidney disease).

### 2.3. Types of Interventions

The interventions included in this review included the following interventions conducted in any settings including schools or communities:Dietary interventions including nutrition education and provision of balanced meals;Physical activity including promotion of physical exercise and reduction in sedentary behaviors;Behavioral therapy;Combination of any of these interventions.

Single interventions were analyzed separately from those studies, where a combination of interventions was being used. The control group comprised of no intervention or standard of care (whatever is applicable in the study setting). We did not include studies comparing any two of the above mentioned intervention. We only included studies with a minimum duration of 12 weeks for intervention and follow-up.

### 2.4. Types of Outcome Measures

The primary outcomes of this study included BMI *z*-scores, BMI, change in body weight, and adverse events (including symptoms associated with low calorie diet). The secondary outcomes included prevalence of overweight and obesity, percentage body fat change, skin fold thickness, waist circumference, health-related quality of life, self-esteem, and cost effectiveness of the intervention. Explanatory secondary outcomes included intensity of physical activity and total caloric consumption. We included studies that reported either subjectively measured or self/-parent-reported outcomes.

### 2.5. Search Methods

We used a comprehensive search strategy to identify eligible studies in Cochrane Controlled Trials Register (CENTRAL), MEDLINE, EMBASE, CINAHL, PsycINFO, the World Health Organization (WHO) nutrition databases (http://www.who.int/nutrition/databases/en/), Social Science Index, and Dissertation Abstracts International. The trials registry (Clinicaltrials.gov) was also searched for ongoing trials. We searched Google along with key nutrition agencies databases, such as Nutrition International, the Global Alliance for Improved Nutrition, the World Food Program, and HarvestPlus, to search for non-indexed, grey literature to locate relevant program evaluations. We also screened the reference lists of all included studies and relevant reviews to identify any additional trials that were not found by the electronic searches. We did not apply any restrictions based on publication, language, or publication status. We included studies published in or after 1990. The search was conducted up to February 2020.

### 2.6. Data Collection and Analysis

Two pairs of review authors (ZAP and AYS; ZH and RAS) independently assessed potential study eligibility using predefined screening criteria. We retrieved the full text of all studies which passed this first level of screening. Disagreements were resolved through discussion with third review author (JKD and RAS) until a consensus was reached. Reasons for exclusion of studies was documented. We extracted data on the study background (time and country where the study was conducted), description of study participants, study design, description of study arms, sample size in each arm, baseline characteristics of the groups, description of intervention, and control groups along with the primary and secondary outcomes listed above.

We performed statistical analysis using RevMan 5 [34]. The included studies were classified as either prevention or management studies and then further classified according to the type of intervention and then analyzed accordingly. We analyzed dichotomous data using risk ratio (RR) with 95% confidence intervals (CI) while for continuous data, we used the mean difference (MD) with 95% CI. We used the standardized mean difference (SMD) with 95% CI to combine trials that measured the same outcome, but used different methods of measurement. We assessed heterogeneity among studies in population interventions or outcomes visually using forest plots. Secondly, heterogeneity between trial results were tested using a standard Chi^2^ test, to assess whether observed differences in results are compatible with chance alone [35]. We reported statistical heterogeneity as I^2^, Q, and tau^2^ for all random-effects meta-analyses. We conducted separate meta-analysis based on the studies assessing obesity prevention and obesity management and type of interventions (dietary intervention, physical activity, behavioral therapy, or any combination of these) within these comparisons. We anticipated heterogeneity within included studies and also performed sensitivity analysis based on study quality. Sensitivity analysis was conducted for all the primary outcomes by removing the studies judged to be at high or unclear risk of bias for sequence generation and incomplete outcome data and the estimates were reported with and without sensitivity analysis.

### 2.7. Quality Assessment

Two pairs of review authors (ZH and ZAP; and AYS and RAS) independently assessed quality of studies, and risk of bias for each study. For randomized studies, we used the Cochrane Risk of Bias tool recommended by the Cochrane Handbook for Systematic Reviews of Interventions [35], which assesses selection, performance, detection, attrition, and reporting bias. Each component was rated as ‘high’, ‘low’, or ‘unclear’ risk of bias. For non-randomized studies, we used the Cochrane Effective Practice and Organization of Care (EPOC) risk of bias criteria (based on additional criteria including baseline characteristics, outcome measurements, protection against contamination, intervention independent of other changes, shape of intervention effect pre-specified, and intervention unlikely to affect data collection) and rated the studies as low risk, high risk, or unclear risk [36]. The quality of evidence was summarized according to the outcomes as per Grading of Recommendations, Assessment, Development and Evaluation (GRADE) criteria [37]. Grades of ‘high’, ‘moderate’, ‘low’, and ‘very low’ were used for grading the overall evidence [38]. For non-randomized studies, the quality of evidence was updated based on large magnitude of effect, dose response, and effect of all plausible confounding factors reducing, which suggests a spurious effect. Two reviewers discussed ratings, reached consensus, and disagreements were resolved by consulting a third reviewer. We developed a summary of findings table to show the effects for the primary outcomes.

## 3. Results

### 3.1. Results of the Search

The search identified 40,775 records from outlined search engines. After the removal of 5446 duplicates, the remaining 35,329 records underwent title and abstract screening. We included 654 studies (1160 papers) for data extraction, and meta-analysis (see Figure 1 for study flow diagram). The major reasons for excluding studies from full text screening stage included wrong study design, wrong comparison group and duration of intervention being <12 weeks.

### 3.2. Description of Included Studies

Out of the 654 included studies, a total of 359 studies focused on obesity prevention; a total of 280 studies targeted obesity management, while 15 studies targeted both prevention and management. The following interventions were assessed for obesity prevention: Diet only interventions—34 studies;Exercise only interventions—57 studies;Behavioral therapy only intervention—89 studies;Diet and Exercise interventions—99 studies;Diet and Behavioral therapy interventions—7 studies;Exercise and behavioral therapy interventions—47 studies;Diet, exercise and behavioral therapy interventions—26 studies.

The following interventions were assessed for obesity management:Diet only interventions—17 studies;Exercise only interventions—59 studies;Behavioral therapy only intervention—63 studies;Diet and Exercise interventions—57 studies;Diet and Behavioral therapy interventions—5 studies;Exercise and behavioral therapy interventions—30 studies;Diet, exercise and behavioral therapy interventions—49 studies.

Among the included studies, a total of 515 studies were RCTs and 139 studies were quasi-experimental studies. Majority of the included studies (*n* = 531, 81.1%) were conducted in HICs, while 70 studies (10.6%) studies were conducted in Upper Middle Income Countries (UMICs) and 13 studies (2%) were conducted in LMICs. Characteristics of all the included studies are summarized in Appendix A.

### 3.3. Risk of Bias

#### 3.3.1. For Randomized Control Trials

The Cochrane Risk of Bias tool was used to assess the quality of the included RCTs. Six studies were judged to be at high risk of bias for random sequence generation, while 221 studies were judged to be at unclear risk of bias since these studies did not specify the methods used to generate random sequence. A total of 288 studies were judged to be at low risk of bias for sequence generation, since appropriate methods were used to generate a random sequence. Thirteen studies were judged to be at high risk of bias for allocation concealment since they did not adequately conceal the allocation, while 423 studies did not specify the methods used to conceal allocation and hence were judged to be at unclear risk of bias. A total of 79 were judged to be at low risk of bias for allocation concealment since adequate methods were used to conceal the allocation. Blinding of participants and personnel was adequately done in 81 studies. A total of 162 studies were judged to be at high risk of blinding of participants and personnel, since no blinding was done, while 272 studies failed to report on blinding of participants and personnel. Blinding of outcome assessors was successfully done in 103 studies; however, 89 studies failed to blind outcome assessors and 323 studies did not report on blinding of the outcome assessors and hence were judged to be at unclear risk. A total of 314 studies were judged to be at low risk of bias for incomplete outcome data, while 142 studies had high attrition rates and were judged to be at high risk of bias for attrition. Attrition could not be calculated due to insufficient information in 59 studies and hence were judged to be at unclear risk of bias. A total of 211 studies were at low risk for selective reporting since they had referred to the pre-published protocol or trial registration details, while 304 studies were judged to be at unclear risk for selective reporting since it could not be assessed due to lack of availability of pre-published protocol or trial registration number. The majority of the studies (*n* = 496) were judged to be at low risk for any other bias while 19 studies were judged to be at high risk for other biases. Figure 2 depicts the summary risk of bias graph for the included RCTs.

#### 3.3.2. For Quasi-Experimental Studies

The EPOC criteria was used to assess the quality of the quasi-experimental studies. All the included quasi-experimental studies were judged to be at high risk for sequence generation. A total of 38 studies were judged to be at high risk of bias for allocation concealment while insufficient information did not permit judgement in 101 studies. Baseline outcome measurements were similar across groups in 73 studies. A total of 32 studies were judged to be at high risk for similarity in baseline measurements while there was insufficient data in 34 studies and hence were judged to be at unclear risk of bias. Baseline characteristic were similar across groups in 77 studies, while in 27 studies the baseline characteristics were not similar across the groups and hence were labelled as high risk. There was insufficient information in 35 studies regarding baseline characteristics and hence were judged to be at unclear risk. A total of 68 studies were judged to be at low risk of bias for attrition while 34 studies did not report on attrition and were judged to be at unclear risk. The attrition rate was high in 37 studies and were judged to be at high risk of bias. Knowledge of allocation was adequately prevented in 11 studies; it was not prevented in 39 studies, while in 89 studies there was insufficient information and hence these were judged to be at unclear risk. A total of 31 studies were adequately protected from contamination; 11 studies did not prevent contamination between groups while 97 studies did not provide sufficient information on protection against contamination. Majority of the quasi-experimental studies (*n* = 121) were judged to be at unclear risk for selective reporting due to lack of published protocols or study registration numbers while 18 studies were judged to be at low risk for selective reporting. A total of 135 studies were free from other risk of bias while only four studies were judged to be at high risk for any other biases. Figure 3 depicts the summary risk of bias graph for the included quasi-experimental studies.

### 3.4. PROGRESS Findings

The findings from the PROGRESS are summarized in Table 1:

### 3.5. Effects of Interventions

#### 3.5.1. Comparison 1: Obesity Prevention

A total of 359 studies focused on obesity prevention. Studies reported primary outcomes including studies reported BMI, BMI *z*-score, and body weight. None of the included studies reported any adverse events. Among secondary outcomes, included studies reported prevalence of overweight/obesity, percentage body fat change, skin fold thickness, waist and hip circumference, health related quality of life, and cost effectiveness.

##### Primary Outcomes

The analysis shows that combined diet and exercise interventions might reduce BMI *z*-score (MD: −0.12; 95% CI: −0.18 to −0.06; 32 studies; 33,039 participants; I^2^ 93%; low quality evidence; Figure 4), BMI by 0.41 kg/m^2^ (MD: −0.41 kg/m^2^; 95% CI: −0.60 to −0.21; 35 studies; 47, 499 participants; I^2^ 98%; low quality evidence) and bodyweight by 1.59 kg (MD: −1.59; 95% CI: −2.95 to −0.23; 17 studies; 35,023 participants; I^2^ 100%; low quality evidence; Figure 5) when compared to control group. The BMI *z*-score might also reduce by behavioral therapy only (MD: −0.07; 95% CI: −0.14 to −0.00; 19 studies; 8569 participants; I^2^ 76%; low quality evidence; Figure 6); combined exercise and behavioral therapy (MD: −0.08; 95% CI: −0.16 to −0.00; 9 studies; 7334 participants; I^2^ 74%; low quality evidence), and diet in combination with exercise and behavioral therapy (MD: −0.13; 95% CI: −0.25 to −0.01; 5 studies; 1806 participants; I^2^ 62%; low quality evidence; Figure 7) when compared to the control group.

There was no difference of effect of diet only and exercise only interventions on BMI *z*-score, BMI, or bodyweight when compared to the control group. Diet in combination with behavioral therapy intervention had no difference on BMI *z*-score when compared to the control group, while there was no difference in effect of behavioral therapy alone, exercise in combination with behavioral therapy intervention, and diet in combination with exercise and behavioral therapy intervention on BMI when compared to the control group.

##### Sensitivity Analysis for the Primary Outcomes

In order to explore the heterogeneity, we conducted sensitivity analysis by removing the studies that were quasi-experimental or at high risk or unclear risk of bias for sequence generation and incomplete outcome data. There was no changes in the estimates for the outcome of BMI in the sensitivity analysis. However, the previous effect of diet in combination with exercise and behavioral therapy on BMI *z*-score disappeared after removing studies at high risk of bias or quasi experimental studies (MD: −0.16; 95% CI: −0.36, 0.04; 3 studies; 644 participants; I^2^ 72%). There was asignificant reduction for behavioral therapy alone on body weight by 0.8 kg after removing studies at high risk of bias or quasi experimental studies (MD: −0.80; 95% CI: −1.57, −0.04; 4 studies; 931 participants; I^2^ 76%).

##### Secondary Outcomes

Combined diet and exercise interventions might reduce percentage of body fat by 0.95 kg compared to controls (MD: −0.95; 95% CI: −1.28 to −0.61; 10 studies; 19,643 participants; I^2^ 80%; low quality evidence). Combined exercise and behavioral therapy might reduce the skinfold thickness (triceps) (MD: −1.33; 95% CI −1.89 to −0.76; five studies; 2944 participants) and increase physical activity intensity (MD: 0.84; 95% CI: 0.09 to 1.59; 14 studies; 5924 participants). We are uncertain of the effect of any of the obesity prevention interventions on any of the other secondary outcomes including prevalence of overweight, prevalence of obesity, waist circumference and health-related quality of life (Appendix A).

A few of the included studies assessed cost effectiveness of the interventions for obesity prevention, mainly in the school settings. The cost-effectiveness study of the “Join the Healthy Boat” program evaluating state-wide implementation of the health promotion in primary schools in Germany [39] suggested that the positive impacts of the study were achieved at affordable costs and with proven cost-effectiveness. The findings from the economic evaluation of the Physical Activity 4 Everyone (PA4E1) intervention [40], which was a multi-component intervention implemented in secondary schools located in low-income communities also suggested that PA4E1 was a cost effective intervention for increasing the physical activity levels and reducing unhealthy weight gain in adolescence. The cost effectiveness analysis of the CHIRPY DRAGON study suggested that this school and family based obesity prevention programme was not only effective but highly cost effective in reducing BMI z scores in primary-school–aged children in China [41]. The economic evaluation of the Healthy Caregivers-Healthy Children (HC2) program, which is an early childcare center-based obesity prevention program, suggested that the HC2 intervention shows potential for generating cost savings [42]. One study (the WAVES study) [43] assessing the cost-effectiveness of a multi-faceted school-based obesity prevention intervention targeting children aged 6–7 years suggested that more research to explore obesity prevention within schools as part of a wider systems approach to obesity prevention are needed. One study also assessed the cost-effectiveness of a large, multifaceted, community-based capacity-building demonstration program (Be Active Eat Well (BAEW)) and suggested that BAEW was affordable and cost-effective [44].

#### 3.5.2. Comparison 2: Obesity Management

A total of 280 studies assessed obesity management interventions. Studies reported primary outcomes including BMI, BMI *z*-score, and body weight. None of the included studies reported any adverse events. Among secondary outcomes, included studies reported prevalence of overweight/obesity, percentage body fat change, skin fold thickness, waist and hip circumference, health related quality of life, and cost effectiveness.

##### Primary Outcomes

The analysis shows that exercise only interventions probably reduces BMI *z*-score (MD: −0.13; 95% CI: −0.20 to −0.06; 12 studies; 1084 participants; I^2^ 0%; moderate quality evidence; Figure 8), and might reduce BMI (MD: −0.88; 95% CI: −1.26 to −0.50; 34 studies; 3846 participants; I^2^ 72%; Figure 9) and body weight (MD: −3.01; 95% CI: −5.56 to −0.47; 16 studies; 1701 participants; I^2^ 78%; low quality evidence; Figure 10) when compared to the control group. The exercise along with behavioral therapy interventions (MD: −0.08; 95% CI: −0.16 to −0.00; 8 studies; 466 participants; I^2^ 49%; moderate quality evidence; Figure 11), diet along with behavioral therapy interventions (MD: −0.16; 95% CI: −0.26 to −0.07; 4 studies; 329 participants; I^2^ 0%; moderate quality evidence; Figure 12) and combination of diet, exercise and behavioral therapy (MD: −0.09; 95% CI: −0.14 to −0.05; 13 studies; 2995 participants; I^2^ 12%; moderate quality evidence; Figure 13) also probably decreases BMI *z*-score when compared to the control group. There was no difference of effect of diet only or diet in combination with exercise on BMI *z*-score when compared to the control group 9 (Appendix A). Behavioral therapy alone might reduce BMI (MD: −0.44; 95% CI: −0.78 to −0.11; 26 studies; 3642 participants; I^2^ 82%; Figure 14), and a combination of diet and exercise interventions might reduce the bodyweight (MD: −2.07; 95% CI: −2.90 to −1.24; 24 studies; 4415 participants; I^2^ 86%; low quality evidence; Figure 15).

There is no difference of effect of diet only, exercise in combination with behavioral therapy, diet in combination with behavioral therapy, or diet in combination with exercise and behavioral therapy on BMI and bodyweight when compared to the control group. There was also no difference of effect of diet in combination with exercise on BMI and behavioral therapy alone on bodyweight when compared to the control group.

##### Sensitivity Analysis for the Primary Outcomes

In order to explore the heterogeneity, we conducted sensitivity analysis for the primary outcomes by removing the quasi experimental studies and the studies at high or unclear risk of bias for sequence generation and incomplete outcome data. After sensitivity analysis, there was significant reduction in BMI with diet along with exercise interventions which was previously statistically non-significant (MD: −0.5; 95% CI: −0.85 to −0.16; I^2^ 87%). There was a significant reduction in BMI with the combination of diet, exercise, and behavioral therapy interventions (MD: −0.51; 95% CI: −0.89 to −0.13; I^2^ 82%). For BMI *z*-score, diet along with exercise interventions also showed significant effect on BMI *z*-score (previously non-significant) (MD: −0.09; 95% CI: −0.18 to −0.01; I^2^ 91%). There was no change in body weight after sensitivity analysis.

##### Secondary Outcomes

Exercise only interventions might reduce percentage body fat (MD: −1.36; 95% CI: −2.32 to −0.39; 25 studies; 1635 participants; I^2^ 78%). Diet in combination with exercise (MD: −2.03; 95% CI: −3.50 to −0.56; 8 studies; 2234 participants; I^2^ 87%; low quality evidence) and a combination of diet, exercise and behavioral therapy (MD: −1.54; 95% CI: −2.56 to −0.52; 11 studies; 3018 participants; I^2^ 63%; low quality evidence) might reduce waist circumference. Behavioral therapy only might reduce the total caloric consumption (MD: −131.58; 95% CI: −188.16 to −75.01; 6 studies; I^2^ 13%; 919 participants). There was no effect of any intervention on any other secondary outcomes including prevalence of overweight, prevalence of obesity and skin fold thickness (Appendix A).

Few studies reported the cost-effectiveness of obesity prevention interventions. Three studies [45,46,47] compared the cost-effectiveness of Family-Based Group Treatment for Child and Parental Obesity, suggesting that for families with overweight/obese children and parents, family based therapy might be a low cost strategy compared to treating the parent and child separately. Study assessing the economic evaluation of the Families for Health program focusing on a parenting approach, designed to help parents develop their parenting skills to support lifestyle change within the family suggested that the program was neither effective nor cost-effective for the management of obesity in children aged 6–11 years, in comparison with usual care [48]. The economic evaluation of a childhood obesity intervention with electronic decision support for clinicians and self-guided behavior-change support for parents suggested that these interventions might be more cost-effective than previous clinical interventions [49]. An economic evaluation was conducted for the Whānau Pakari [50], which was a home-based, 12-month multi-disciplinary child obesity intervention programme, suggesting that such programs had lower programme costs per child, greater reach, with similar impacts. However, one study assessing the cost-effectiveness of group treatment compared with routine counseling in obese children suggested that family-based group treatment was more costly compared with individual routine counseling and the salaries were the major component of the total costs [51]. One study assessing the cost-utility of a motivational multicomponent lifestyle-modification intervention in a community setting (the Healthy Eating Lifestyle Programme (HELP)) [52] suggested that there was no evidence that the program was more effective than a single educational session in improving quality of life in a sample of adolescents with obesity. A study assessing the cost-effectiveness of an intensive weight-loss intervention for children compared with a low-intensity intervention suggested that, compared with the standard care, the camp group was more costly [53]. The cost effectiveness of the Live, Eat and Play (LEAP) study suggested that the intervention led to higher costs to families and the health care sector, which could have been devoted to other uses that do create benefits to health and/or family well-being [54].

## 4. Discussion

This review was a comprehensive review on obesity prevention and management interventions in children and adolescents from HICs as well as LMICs. The review summarizes findings from a total of 654 studies from 1160 papers. A total of 359 of the included studies focused on obesity prevention while 280 studies focused on obesity management. About 15 studies focused on both treatment and prevention. Majority of the studies (about 81%) were conducted in HIC; about 10% of the included studies were conducted in UMICs while only 2% of the included studies were conducted in LMIC settings. The interventions evaluated for obesity prevention and management included diet only interventions; exercise only interventions; behavioral therapy only interventions; diet along with exercise, diet along with behavioral therapy, exercise along with behavioral therapy and a combination of diet, exercise and behavioral therapy. About 515 studies were RCTs, while 139 studies were quasi-experimental studies. Among RCTs, the majority of the studies were judged to be at low risk of bias for sequence generation and incomplete outcome data while majority of the studies were labelled as unclear risk for blinding and selective reporting due to insufficient information provided. Among quasi-experimental studies, almost all the studies were judged to be at high risk for sequence generation, while the majority of the studies were judged to be at unclear risk for allocation concealment, prevention of knowledge of the intervention, contamination, and selective reporting. Due to the nature of the intervention, majority of the studies could not achieve blinding of the participants and personnel. Overall, the outcomes were judged to be of moderate to low quality due to study limitations, high heterogeneity. and imprecision. Among the PROGRESS factors, studies reported on a few factors, including place of residence/setting; gender/sex; education and recruitment methods. The least reported factors were found to be race/ethnicity/culture/language; occupation; religion; socioeconomic status; and social capital.

For primary outcomes, evidence for the prevention of obesity among children and adolescents suggests that a combination of diet and exercise might reduce the BMI, BMI *z*-score and body weight. Behavioral therapy alone and a combination of exercise and behavioral therapy might reduce BMI *z*-score. Sensitivity analysis suggested that behavioral therapy might also reduce body weight. For the secondary outcomes, combined diet and exercise interventions might reduce percentage of body fat while combined exercise and behavioral therapy might reduce the skinfold thickness (triceps) and increase physical activity intensity. There was no effect of any other intervention on any of the other outcomes.

For obesity management, evidence suggests that Exercise only interventions, diet along with behavioral therapy interventions and a combination of diet, exercise and behavioral therapy probably decreases BMI *z*-score. Sensitivity analysis suggests that diet along with exercise might also reduce BMI *z*-score. Exercise only and behavioral therapy only might reduce BMI while the sensitivity analysis suggests that diet along with exercise and a combination of diet, exercise and behavioral therapy might also reduce BMI in overweight/obese children and adolescents. Exercise only interventions and diet along with exercise might reduce body weight. There was no effect of any other intervention on any of the other outcomes.

Despite there being several isolated and combination of interventions available for addressing childhood and adolescent obesity, this review demonstrated that in dealing with obesity might require a combination of interventions rather than each intervention on its own. Primary prevention of obesity in children and adolescents has been a major public health challenge for several decades [55]. Lifestyle modifications form the mainstay of primary prevention for this condition [55]. The current review highlighted that combination of dietary and exercise interventions might led to significant reductions in BMI, BMI z score and body weight. This change is likely mediated by increase in energy expenditure as indicated by the increase in physical activity intensity while there was no change in the total caloric consumption. Vissers et al. also reported that diet and exercise interventions had a greater impact on adipose tissue as compared to diet alone [56], thus indicating the role of energy expenditure in weight loss.

For obesity management, lifestyle modification along with pharmacological and surgical interventions have been proposed; however the focus of this review was limited to lifestyle modifications only. A recent review of 8 trials on approximately 4000 obese children indicated that diet and exercise interventions for 6 months helped reduced BMI z scores and other markers of metabolic dysfunction such as fasting plasma glucose in these children [57]. Our findings indicate that adding behavioral modifications such as family based therapy or cognitive behavioral therapy (CBT) to dietary and exercise interventions might have an impact on BMI and skinfold thickness along with BMI z scores. Family based therapy where parents and children are actively involved in making healthier nutrition and physical activity choices is one of the most robust interventions for childhood obesity [58]. Complementing family based therapy with CBT where individuals are encouraged to change attitudes and behaviors that sustain a current behavior have also proven to be beneficial in this age group [59]. Thus, a comprehensive “body and mind” intervention may result in a greater impact on improving body composition [60].

This review serves as an umbrella review of the existing evidence for childhood and adolescent obesity prevention and management. It synthesizes global evidence from the RCTs and quasi-experimental studies along with the PROGRESS factors. This review included anthropometric as well as other outcomes such as health related quality of life and cost effectiveness of the interventions. However, this review only provides a basis for what works and what does not work for childhood and adolescent obesity. Future reviews in the domain should further microscopically evaluate these findings in terms of effectiveness in specific subgroups of population and types of intervention. Childhood obesity is a known risk factor for adult obesity and other non-communicable diseases leading to morbidity and mortality. This may potentially lead to loss of human capital and economic challenges at the family and community level [61]. Future work in this domain should assess the impact of these interventions on outcomes other than anthropometry like quality of life and health economic evaluations to better inform policy makers [61]. There was a significant dearth of literature on obesity management and prevention from LMICs. This lack of evidence continues to exist despite the well documented rise of obesity in LMICs, hence, the findings from this review may have limited applicability in such regions.

## 5. Conclusions

The existing evidence is most favorable for a combination of interventions, such as diet and exercise in obesity prevention and diet, exercise, and behavioral therapy for obesity management. Despite the growing obesity epidemic in LMICs, there is significant dearth of obesity prevention and management studies from these regions. Future studies in this domain should focus on combinations of interventions with an appropriate follow up period to generate robust evidence on weight and related parameters, with evidence from LMICs. 

## Figures and Tables

**Figure 1 nutrients-12-02208-f001:**
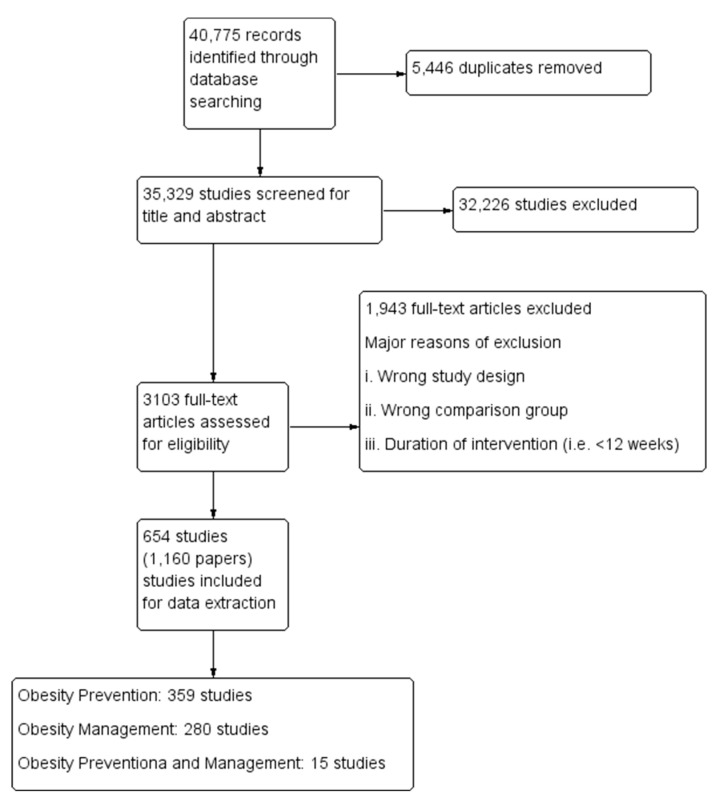
Study flow diagram.

**Figure 2 nutrients-12-02208-f002:**
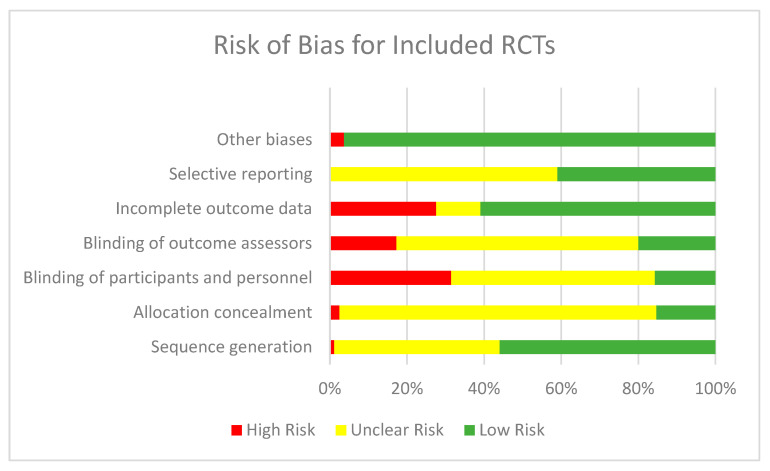
Summary risk of bias for included randomized controlled trials (RCTs).

**Figure 3 nutrients-12-02208-f003:**
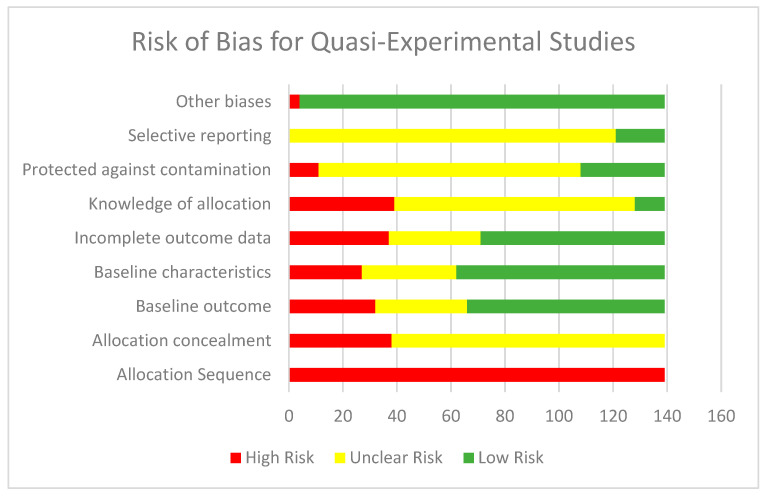
Summary risk of bias for included quasi-experimental studies.

**Figure 4 nutrients-12-02208-f004:**
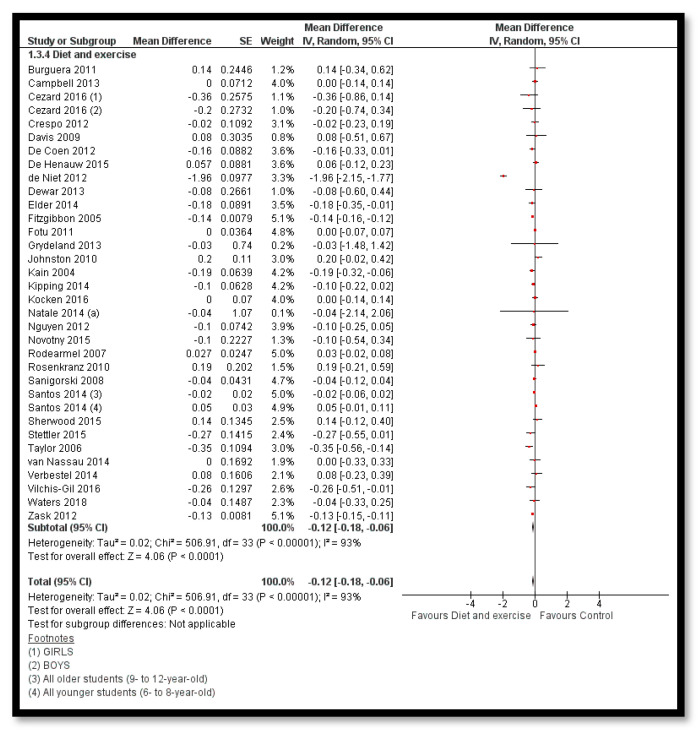
Forest plot for the effect of combined diet and exercise interventions for obesity prevention on body mass index (BMI) *z*-score.

**Figure 5 nutrients-12-02208-f005:**
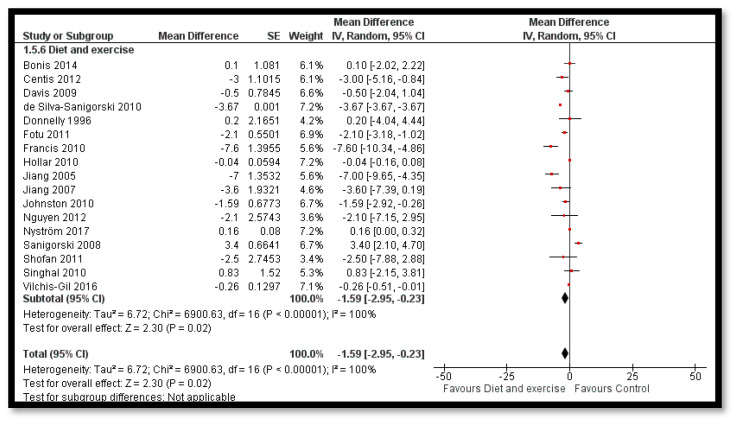
Forest plot for the effect combined diet and exercise interventions for obesity prevention on body weight.

**Figure 6 nutrients-12-02208-f006:**
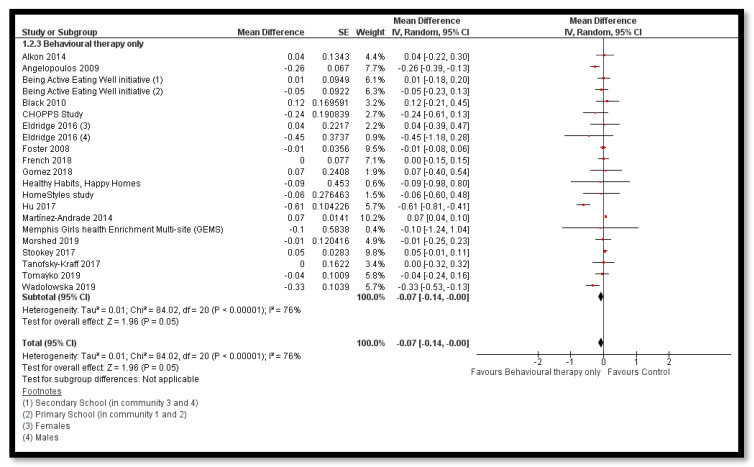
Forest plot for the effect of behavioural therapy interventions for obesity prevention on body mass index (BMI) *z*-score.

**Figure 7 nutrients-12-02208-f007:**
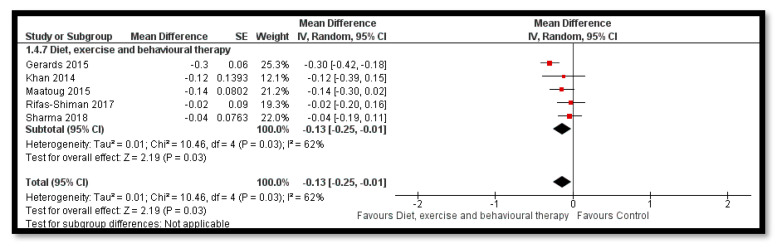
Forest plot for the effect of combined diet, exercise and behavioral therapy interventions for obesity prevention on body mass index (BMI) *z*-score.

**Figure 8 nutrients-12-02208-f008:**
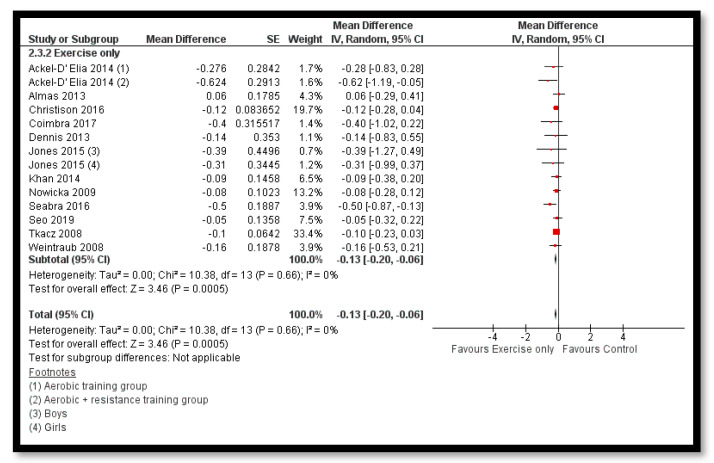
Forest plot for the effect of exercise only interventions for obesity management on body mass index (BMI) *z*-score.

**Figure 9 nutrients-12-02208-f009:**
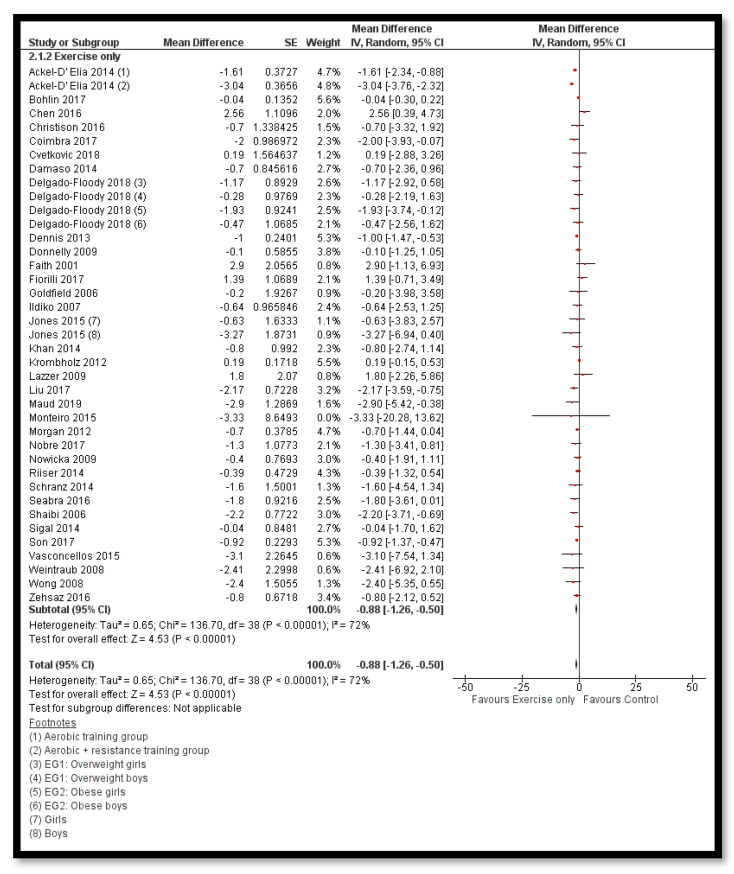
Forest plot for the effect of exercise only interventions for obesity management on body mass index (BMI).

**Figure 10 nutrients-12-02208-f010:**
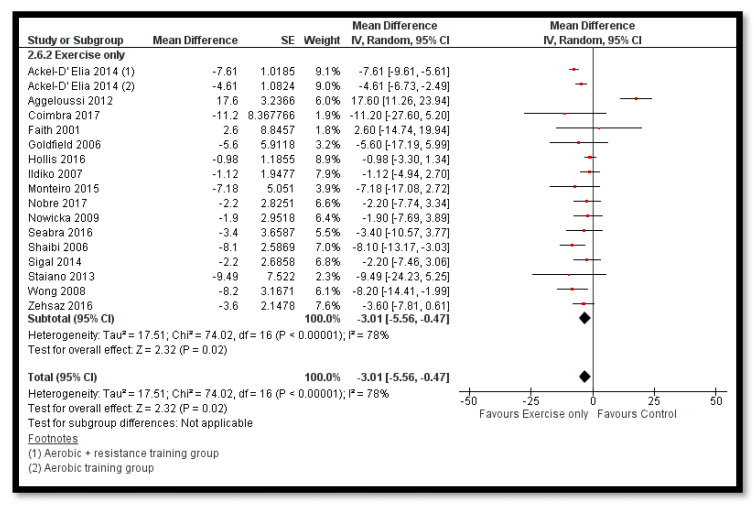
Forest plot for the effect of exercise only interventions for obesity management on body weight.

**Figure 11 nutrients-12-02208-f011:**
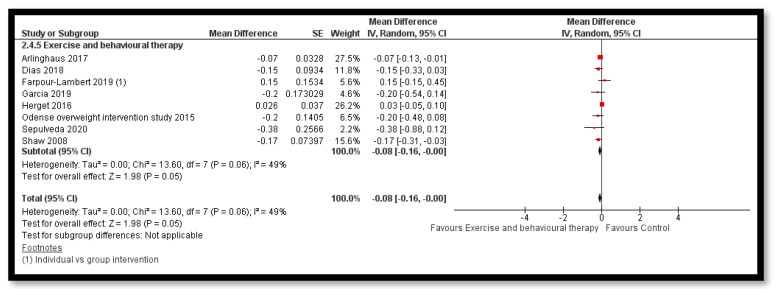
Forest plot for the effect of exercise and behavioural therapy interventions for obesity management on body mass index (BMI) *z*-score.

**Figure 12 nutrients-12-02208-f012:**
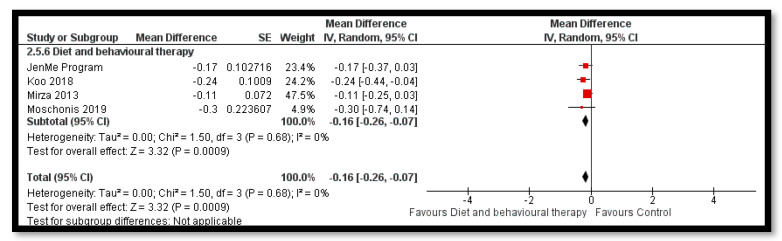
Forest plot for the effect of diet and behavioural therapy interventions for obesity management on body mass index (BMI) *z*-score.

**Figure 13 nutrients-12-02208-f013:**
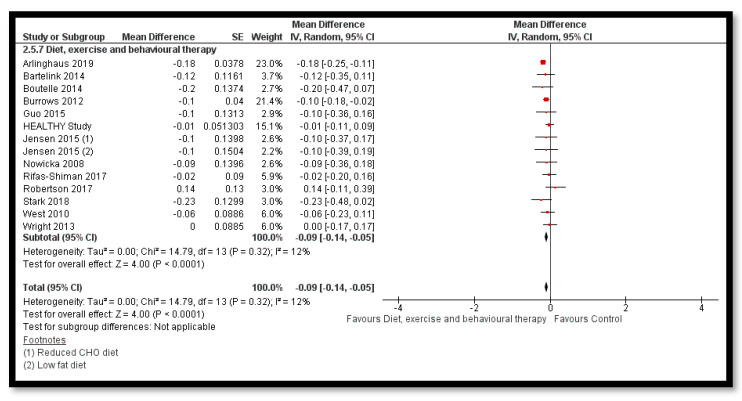
Forest plot for the effect of diet, exercise and behavioural therapy interventions for obesity management on body mass index (BMI) *z*-score.

**Figure 14 nutrients-12-02208-f014:**
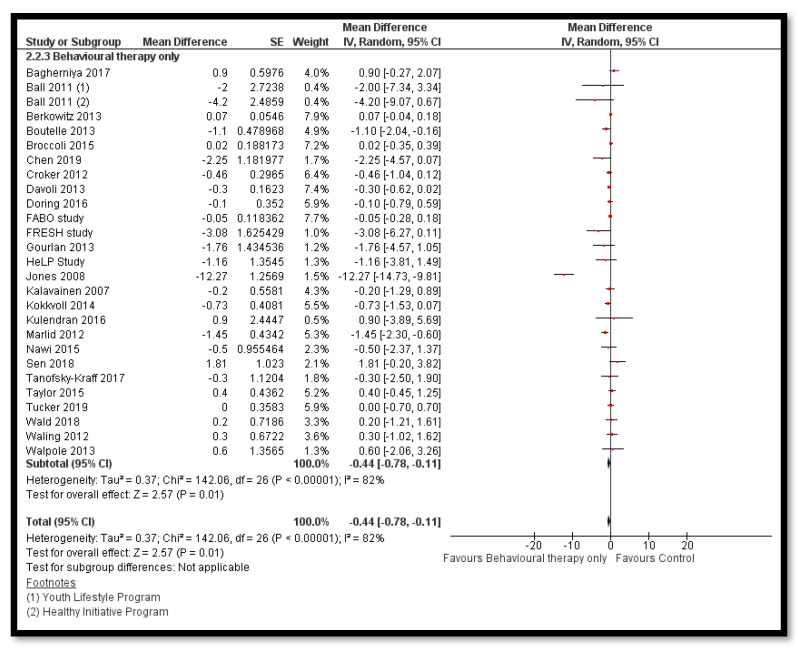
Forest plot for the effect of behavioural therpy only interventions for obesity management on body mass index (BMI).

**Figure 15 nutrients-12-02208-f015:**
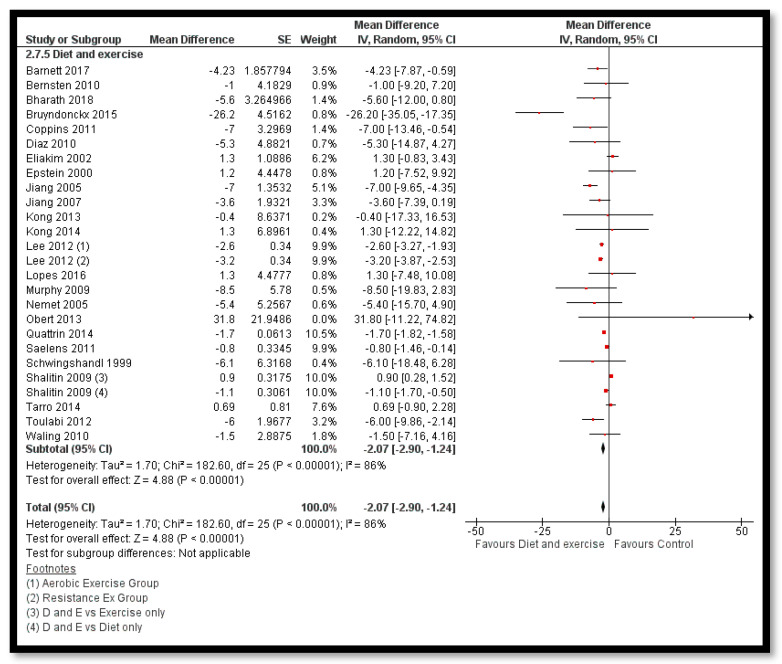
Forest plot for the effect of diet and exercise interventions for obesity management on body weight.

**Table 1 nutrients-12-02208-t001:** The summary of findings from the PROGRESS.

PROGRESS-Plus Factors	Summary of Reported Factors
Place of residence/setting	No. studies conducted in high-income countries (HICs): 533 No. studies conducted in upper-middle-income countries (UMICs): 70 No. studies conducted in lower-middle-income countries (LMICs): 13 40 studies had no country stated.
Race/ethnicity/culture/language	Almost half of the included studies specified details under this domain while reporting the baseline characteristics of the study population.
Occupation	This is one of the most under-reported categories in the studies probably since the study population were children and adolescents. Very few studies reported the occupation of the parents of the enrolled chidlren and adolescents.
Gender/sex	This category was reported in almost every study, although few did not provide the specific distribution of the sample by sex when participants from both sexes were included. Most studies were conducted with children and adolescents, but some also included only adolescents or only children.
Religion	This is also one of the under-reported categories in the studies. Very few studies reported on this domain.
Education	Since many studies were carried out in school settings, majority of such studies reported the level of education as preschool/elementary, primary or secondary.
Socioeconomic status	This factor was also poorly reported in the published data of the included studies. Moreover, various studies used different definitions of the socio-economic status. Majorly, the studies reported income, class, or the areas of residence (rural/urban/mixed) under this domain.
Social capital	Few studies directly reported any measurement of social capital. Indirectly, some studies reported that participants were recruited through schools, clinics, hospitals and sports/recreation centres, thus indicating that participants had at least one social connection or network.
Plus (other characteristics)	All studies reported on age, as this factor is essential for their analysis. Many reported the participants’ Body Mass Index (BMI) and other body measures e.g., height, weight, skinfold thickness. Studies including parents also reported parent education, occupation, income and marital status although very infrequently.
Recruitment methods	Most studies recruited their participants through similar strategies: schools, mailings, printed ads and flyers distributed in school campuses, community centres, clinics or hospitals, through advertisement on local radio and television. Most of the studies took place in HICs and in children and/or adolescents, hence the use of schools and community centres.

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
