# Peer review of "Effects of Lifestyle Modification Interventions to Prevent and Manage Child and Adolescent Obesity: A Systematic Review and Meta-Analysis"

_nutrients, 2020, doi:10.3390/nu12082208_

Round 1
Reviewer 1 Report
I am happy with by revisions you have made, well done and congratulations!
Author Response
Thank you so much for your initial feedback.
Reviewer 2 Report
Salam et al. present an ambitious review and meta-analysis of interventions to prevent and treat child and adolescent obesity. The revised version did not address all the critiques from prior review.
The abstract needs rewriting to be better focused and structured in order to compare effects of dietary, behavioral and exercise interventions. For instance, they write in line 49 that behavioral therapy alone and a combination of exercise and behavioral therapy might reduce BMI z-score. Then in line 52 they write that sensitivity analysis suggested that behavioral therapy might reduce also BMI z-score, and in line 56 they write again that behavioral therapy only …..and so one. This is difficult to read. There are too many details of testing in the abstract.
Data presentation: Looking at changes in body weight in growing children of different age groups and different sexes are meaningless. I would exclude these. Showing changes of BMI is for the same reasons not very helpful. I think this paper needs to be focused on comparing % changes in BMI and comparing changes of BMI z-scores between the different studies.
Fig. 4 can be omitted.
Figures 4-14, the Forest blots are difficult to read, there are no scales and no headers/titles of columns. I recommend changing the order in a meaningful way, for instance showing first data for single interventions and then for combined intervention.
In Fig. 8, please check results from Monteiro 2015.
In Fig. 9, please check result Ball 2011.
In Figure 10, results from Backlund 2011, that must be an error.
Fig. 11, please recheck results from Epstein 2000.
Figs. 13, 14, it is unclear what is shown.
Author Response
Please see the attachment

This manuscript is a resubmission of an earlier submission. The following is a list of the peer review reports and author responses from that submission.
Round 1
Reviewer 1 Report
Salam et al. present an ambitious review and meta-analysis of interventions to prevent and treat child and adolescent obesity. While there are several meta-analyses regarding this topic in the literature, this is a very comprehensive large study. The report is timely and important. There are several problems that need to be addressed. First, relevant information about each selected study should be given, for instance number of subjects, sex, baseline BMI z-score, age range, location. While there is some information how the papers/studies were selected, it is difficult to follow how the data were analyzed. Second, presenting changes of BMI in abstract and Forest plots are meaningless in growing children. Better would be to report % changes in BMI or just focus on reporting BMI z-scores. Therefore, Figures 2 and 5 can be omitted. Third, the presented forest plots are hard to read, fonts and symbols need to be much bigger. I recommend breaking Fig. 3 into two figures, one for single interventions, the other for combination treatments.
Other points:
- Lines 195-201, number of papers and number of references do not match.
- Data on Fig. 4 are unclear, it does not look like BMI z-score data. Please check.
- Lines 312-317, skinfold data as reported are very unclear and difficult to understand. More details and clarity are required.
- Please check for typos and punctuation errors, I found several.
Reviewer 2 Report
General comments
This study attempted to review lifestyle interventions for preventing and managing childhood and adolescent obesity without language restrictions. Although I appreciate the substantial hard work done by the authors, I am concerned of the search strategies used (major intervention studies published in the recent years were missed and this has made the conclusion on research gaps invalid). I also think the manuscript could benefit from more work. Specific comment/suggestions for each section are provided below.
Specific comments
Abstract
- Key information on inclusion/exclusion criteria was missing (e.g. did you include non-controlled trials studies; what were the primary and secondary outcomes and did you include studies using self-reported/subjective measures of adiposity/health behaviours?). This information is essential for readers to judge the strength of the review findings.
- The abstract only provides descriptive results and doesn’t give interpretive conclusions regarding effectiveness of interventions.
- It would have been more helpful to briefly explain in the abstract why this review was needed (i.e. its importance or new contributions – for example how was it different from the latest Cochrane review published in 2019)
- It’s unclear in the abstract how prevention, treatment and population-level studies were handled in this review (prevention and treatment interventions are different in many ways).
- Lines 25-29: It would be more useful to report the number of different types of interventions for prevention-, treatment- and population-based studies separately. The current evidence base for prevention and treatment interventions are not the same.
- Quality of evidence are mentioned but we don’t know how quality was assessed.
- Please check the statement/95% CI (Lines 34-35) : ‘…combined diet and exercise interventions might reduce BMI by 0.41 kg/m2 when compared to control (MD: -0.41 kg/m2; 95% CI: -0.60 to 0.22…)’
Introduction
- More recent references for international prevalence data could have been used. I would recommend the 2019 Global Nutrition Report.
- References for important points (arguments about previous reviews) were missing- lines 90-93.
- Studies cited between Lines 72 and 89 may not provide a balanced overview of the current literature. For a systematic review that attempted to include both prevention and treatment studies without language restrictions, a critical discussion of published, major international reviews of prevention and treatment studies would have been more helpful to put this review into context and explain what this new review was aimed to add. The authors did attempt to discuss previous reviews in lines 93-96 but this was too brief and lacked useful details. For example, Line 95: limited to what factors and how this review addressed these issues? You also said ‘the majority of the existing systematic reviews have restricted their inclusions to randomized controlled trials (RCTs) alone’. Why could this be a limitation instead of a strength? Why shall we include non-controlled studies? I understand the weaknesses of RCT and the methodological issues of trials in LMICs, but most readers would benefit from more information.
Methods
- Line 141: you said the review included studies published in or after 1990. It’s unclear when (date) the search was ended. By looking at the results, major trials published recently were not included.
- Outcome measures – did you include studies using subjectively measured or self/-parent-reported outcomes?
- Could you please explain why BMI was included as a primary outcome? BMI is age and sex dependent.
- This review included studies in various countries/ethnicity groups, and BMI is ethnicity dependent too. Did you have any restrictions on reference/classification systems used, particularly for meta analyses?
- Meta analyses (lines 159-161): You said ‘you conducted separate meta-
- analysis based on the interventions (dietary intervention, physical activity, behavioral therapy, or any combination of these), outcome, and study design (RCT, and quasi experimental studies)’. Does it mean prevention and treatment studies were mixed up in the analyses? However, in the results, it seems that the analyses were separate.
Results
- Line 253 – you said ‘None of the included studies reported cost effectiveness’. To my knowledge, major childhood obesity prevention trials published in the recent years (all in high impact journals) were not included in this review. At least 3 of them conducted a comprehensive cost effectiveness evaluation (although cost effectiveness evaluation results were not always included in the main trial papers – published separately). This may indicate issues in the search strategies (or a need to update the search) and so the review findings.
- Figure 1: Please report reasons (broad categories are fine) of exclusion at full-text assessment stage.
Discussions
- Lines 348-351: you said ‘This review included anthropometric as well as other outcomes such as health related quality of life and cost effectiveness of the interventions. It is important to note that information regarding the latter two outcomes were lacking from the included studies.’
Again, to my knowledge, this statement is incorrect. For example, please read the CHIRPY DRAGON study (a major RCT in a LMIC with quality of life outcomes and cost effectiveness evaluation), the WAVE study (a major RCT in the UK with quality of life outcomes and cost effectiveness evaluation) and the Daily Mile study (in the UK with cost effectiveness evaluation and quality of life outcomes).
- As with the Introduction, it would be helpful to discuss the review’s findings and contributions in relation to published major reviews. The current discussion is a bit thin at the moment.